# The Effectiveness of a Board Game-Based Oral Hygiene Education Program on Oral Hygiene Knowledge and Plaque Index of Adults with Intellectual Disability: A Pilot Study

**DOI:** 10.3390/ijerph18030946

**Published:** 2021-01-22

**Authors:** Hsiu-Yueh Liu, Ping-Ho Chen, Wun-Jyun Chen, Shan-Shan Huang, Jen-Hao Chen, Ching-Teng Yao

**Affiliations:** 1Department of Oral Hygiene, College of Dental Medicine, Kaohsiung Medical University, Kaohsiung 80708, Taiwan; hyliu@kmu.edu.tw (H.-Y.L.); b0804343@cgu.edu.tw (W.-J.C.); 2Department of Medical Research, Kaohsiung Medical University Hospital, Kaohsiung Medical University, Kaohsiung 80708, Taiwan; 3School of Dentistry, College of Dental Medicine, Kaohsiung Medical University, Kaohsiung 80708, Taiwan; phchen@kmu.edu.tw (P.-H.C.); jehach@kmu.edu.tw (J.-H.C.); 4Division of Pediatric Dentistry and Special Care Dentistry, Department of Dentistry, Kaohsiung Medical University Hospital, Kaohsiung 80708, Taiwan; 1070480@kmuh.org.tw; 5Master Program of Long-Term Care in Aging, Kaohsiung Medical University, Kaohsiung 80708, Taiwan

**Keywords:** effectiveness, intellectual disability, board game, oral health related knowledge, plaque index

## Abstract

An oral hygiene board game was designed as an intervention tool and applied to explore the effectiveness for the oral health related knowledge (OHK) score and plaque index (PI) of adults with intellectual disability (ID). This one-group pre/post-test design study was conducted in a residential long-term care facility for 42 participants. The study had one pre-test (baseline) and three post-tests evaluated in both control and intervention stages, respectively. The participants participated in a 60-min oral hygiene board game twice a week during the intervention stage. Total OHK score and PI of the participants were recorded to determine the effectiveness of intervention. There were no differences in OHK score and PI between the two stages at baseline. The results in intervention stage demonstrated a significant gradual increase and reduction in the OHK score when compared with the control stage. A statistically significant improvement in the OHK score and PI (42.29% and −33.28%, respectively) at the end of intervention between two stages was recorded. This study proved a board game is deemed an effective education method applicable to promote the OHK and skills of ID adults.

## 1. Introduction

The intellectual, medical, physical, social, or psychological development of adults with intellectual disability (ID) is considerably slower than that of their similar-aged ordinary counterparts [1,2,3,4]. People with ID experience difficulty in learning and adapting to life and they usually lack appropriate oral hygiene and oral care habits to care for their own oral health [5,6,7,8]. In Taiwan, people with ID have poorer oral health, such as more untreated decayed teeth (3.36), missing teeth (4.86) [9], and a higher gingivitis care needs (81.31%) compared with their similar-aged ordinary counterparts (1.24, 4.50, and 71.31%) [10]. The oral health condition of accumulated poor oral health exacerbates with the degree of disability and age in people with ID [1,2,9,11,12,13]. ID individuals have more missing teeth and require more medical care than ordinary people from 45 to 64 years old (9.78 and 6.34) to over 65 years old (15.00 and 10.28) [9,10]. Although Taiwan’s government provides free oral preventive health services and supports the dental treatment expenses for more than 20 years, the utilization of preventive health services among people with ID tends to be low [11]. Significantly fewer adults with ID (38.2%) accept dental filling to their decayed teeth, compared with those without ID (57.82%) [9,10]. Due to problems such as low-income household status, relevant chronic diseases, low accessibility to medical services, reluctance of dentists to provide treatment, and lack of cooperation in people with disabilities, the tooth filling utilization of people with ID is significantly lower (17.53%) [11,14]. Previous research stated that oral diseases originate from microorganisms, adequate OH and effective removal of dental plaque are critical to safeguarding oral health [8,13,15].

An oral hygiene education intervention program has been recognized as one common method to improve people’s knowledge, attitude, and practice regarding oral health, reduce plaque, gum bleeding, and chance of dental cavities [15,16]. Oral hygiene education is often provided to people with disabilities through the conventional teaching method of lecturing [17,18]. Makuch and Reschke [19] compared the effects of game-based education and conventional lecture education for children’s oral health promotion. This game-based education involves role-playing, picture matching, and a tooth-brushing song, whereas the conventional lecture–based system consists of common noun introduction. Relevant research has confirmed that the game-based oral hygiene intervention is more efficacious in helping children learn the tooth-brushing technique than the presentation of didactic information alone [19,20,21].

Games have gradually been incorporated into teaching activities in recent years, constituting an emerging education strategy. The process of games provides a non-threatening, but competitive learning environment, which enables learners to focus on content and reinforce their learning. Games facilitate intrinsic motivation as well as learning. The fun experienced by learners in the process of a game promotes motivation, makes it interesting, leads to high learning outcome, and builds self-confidence [22]. Board games, as an exercise for the mind and the hands, are applicable to all ages for the development of various disciplines [23]. Board game-based education activities are gradually being applied to health education [24], and numerous studies have examined relevant topics, including motivation, performance [24,25,26], attitude, affective aspect [27], cognitive development, and knowledge comprehension [28,29,30,31,32]. Most of these studies have identified the positive effects of board game education, proving its effectiveness and value as an alternative teaching tool that supports learning [24,28,32].

Studies have shown that people with moderate to severe ID have an IQ lower than 70 [33]. They can still learn to take care of themselves and engage in learning activities through repetitive systematic training [33,34,35]. Conventional teaching methods focus mainly on knowledge transfer, which may make learning tedious and reduce interest in the process. In addition, board games are often used as a teaching medium to improve the academic learning and interpersonal relationships of children with ID [2,19]. However, few studies have looked at the use of board games as an oral hygiene intervention in people with ID. Therefore, in this study, the board game teaching strategy was used to design an oral hygiene education program that meets the learning needs of adults with ID. In this pilot study, the researcher evaluated the feasibility of applying board games to oral hygiene education in adults with ID and assessed the effectiveness of a board game-based oral hygiene education program on the participants’ oral hygiene and plaque index (PI).

## 2. Methods

### 2.1. Study Design

The present study was a pre-experimental design based on one-group pre/post-test design. This study included control and intervention phases for three weeks, respectively and the interval between two phases was one week. In the control phase, participants did not receive any board game intervention, whereas in the intervention phase, participants received a three-week (total six times) board game-based oral hygiene intervention twice a week (on Monday and Friday), with 60 min per session.

Baseline data of the two phases were conducted by collecting the oral health related knowledge (OHK) and PI values prior to administering the control/intervention before the commencement of first data control/intervention time point (T0), respectively. Post-test data collection was conducted on the 7th (T1), 14th (T2), and 21st (T3) days of two phases for analysis. A total of 8 times data were collected during the control and intervention phases.

### 2.2. Study Population

All participants were recruited from one of 5 long-term care facilities for people with ID in Kaohsiung, southern Taiwan. There are 24 facilities for people with disabilities in Kaohsiung, 13 are facilities for ID, five of the 13 are residential long-term care facilities for adults with ID.

Intellectual disability (ID) is evaluated and certified into four levels: mild, moderate, severe, and profound disability. The inclusion criteria for those staying in the facility in Taiwan were aged from 20 to 64 years old, categorized as moderate or above ID or multiple disabilities (MD) according to their disability identification. All the residents with MD, who had ID and with at least one or more co-occurring disabilities such as hearing, vision, voice, speech or limb disability. All 45 participants (mean age 37.29 ± 9.63 years old) qualified to attend this study. Only one of them could not participate in this study due to rehabilitation and two other people left the facility during the research period. Finally, a total of 42 participants completed this study.

### 2.3. Ethical Approval

This study was approved by the Human Experiment and Ethics Committees of Kaohsiung Medical University Hospital (KMUHIRB-SV(II)-20180061) and was conducted with the consent of the long-term care facility. The principal investigator explained the objective and methods of this study to the participants and their parents/guardians and acquired written consent from them. All research data were encoded to ensure the anonymity of the participants and used only for academic research purposes. The participants were permitted to withdraw from a session or to quit the study altogether during the research procedure for any reason without having their rights to receive care affected.

### 2.4. Assessment

In this study, we collected the demographic data (e.g., sex, age, classification, and degree of disability) from their medical history in the facilities. Baseline (on Friday) OHK and PI of both phases were established by collecting the assessments 3 days prior to conducting the board game intervention. The assessments were conducted once every week out of 3 weeks for post-tests. In total, eight OHK and PI records were collected during the 4 control and 4 intervention collection days from each participant. The assessments were conducted by the same researcher of this study to eliminate any intra-subject variability.

#### 2.4.1. Oral Health Knowledge

Considering the poor cognitive and verbal expression ability of people with ID, the assessment was conducted individually with the assistance of picture identification and physical operation. First, the intervention tool of this research is a board game, we considered the use of a paper-and-pencil test for the questions may affect the relaxed and pleasurable atmosphere of the game. Second, paper-and-pencil answers may affect the result of evaluation due to the reading ability of the participants. We used “selecting the correct picture”, answering “true or false” questions and practical demonstrating action as a substitute plan for the paper-and-pencil test.

The OHK scale contains 25 questions divided into three parts to test the participants’ prevention behavior of dental cavities (seven questions), knowledge of tooth-cleaning products (eight questions), and learning of tooth-cleaning skills (10 questions).

In the part 1 questions, the participants need to select the correct picture to respond to the question, such as “What are the tools for cleaning teeth?” The participants need to correctly identify a toothbrush, interdental brush, mouthwash cup, dental floss, toothpaste, or a dental floss stick in the picture. In the part 2 questions, the participants need to correctly answer “true or false” to questions such as “Do we need to regularly visit the dentist to check the health of our teeth every six months?”. In the part 3 questions, the participants need to correctly demonstrate tooth-brush action according to the question hint, such as, “Brush the outside teeth on the bottom”.

One point is allocated for each correct answer given. The total score ranges between 0 and 25, and the higher the score the greater the OHK. We calculated and recorded the total OHK score for each evaluation.

#### 2.4.2. Plaque Index

A plaque control record [36] was employed to measure the PI of the participants. With a cotton swab, a dental plaque display agent was applied to all surfaces (buccal, lingual, and palatal) of the participants’ teeth surfaces. Subsequently, the participants were asked to rinse their mouth with water, and their teeth were checked with a dental disposable mirror and an auxiliary flashlight. Each tooth was divided into six surfaces (mesio-buccal, centro-buccal, disto-buccal, disto-lingual, centro-lingual, and mesio-lingual) and examined. Missing teeth were excluded from the calculation of the PI. The PI score of a participant was calculated and recorded as follows: the total of the tooth surfaces scored with plaque/the number of tooth surfaces scored × 100%.

### 2.5. Intervention

#### 2.5.1. Development of the Board Game

The oral hygiene board game named “Dental Monopoly”, was designed by the researcher and a panel of experts in the fields of board game design, disabilities, and oral health education. The proposed board game was based on “Snakes and Ladders” combined with Monopoly with the OHK and tooth-cleaning skills to be applied in the game. The board game consisted of a board, a die, four playing pieces, 15 chance cards involving questions about OHK, and 10 fate cards involving the demonstration of tooth-cleaning skills. During the game, the players continually encounter challenges, which require them to answer questions about OHK or to demonstrate their tooth-cleaning skills.

#### 2.5.2. Implementation of the Board Game

The board game activities were conducted in a small group setting and the participants were randomly and evenly divided into two groups. Each group was randomly and evenly divided into four small teams. The board game (intervention) was conducted for two sessions, and each game session conducted twice a week for 3 weeks and lasted 60 min for one group (four small teams) in a group activity room of the long-term care facility.

During the game, each group takes turns to roll the die and to move forward according to the number on the die. If a player moves their game piece to the tail of the toothbrush, the player is allowed to move upwards at the toothbrush head and get ahead faster. If a player lands exactly at the top of a cavity of the tooth, he slides his players game piece all the way to the square at the bottom of the root of the tooth and gets a fate card. The players need to imitate the picture on the card to demonstrate their cleaning skills. If a player moves their game piece to the tooth decay devil, he needs to go back two spaces and misses a turn. If a player moves his game piece to biscuits, candy, or cake, he will get a chance card and answer an oral health related question. If a player correctly answers a question, he will move forward one place; otherwise, he will go back one place. If a player moves his game piece to fluoride toothpaste, dental floss sticks, fresh fruit or vegetables, he will move forward one place. The first player who reaches the highest space on the board, 100, wins the game. The members of the team who win the game will get a prize. Figure 1 shows the oral hygiene board game.

### 2.6. Statistical Analysis

First, the participants’ demographic data, OHK, and PI were entered into Microsoft Excel (Microsoft, Redmond, WA, USA). IBM SPSS Statistics 20.0 (IBM, Armonk, NY, USA) was then used to perform descriptive and inferential statistical analysis. Demographic data are expressed as percentages, and OHK and the PI are presented as means and standard deviations. A paired *t*-test was used to compare baseline and post-test results to assess differences in OHK and PI before and after the board game intervention. The level of significance was set at *p* < 0.05.

### 2.7. Ethical Approval

The study was approved by a relevant Ethics Committee of Kaohsiung Medical University Hospital (KMUHIRB-SV(II)-20180061). All procedures performed were in accordance with the ethical standards of the institutional and/or national research committee and with the 1964 Helsinki Declaration and its later amendments or comparable ethical standards. All participants signed the consent form prior to participation.

## 3. Results

As shown in Table 1, the majority of the research participants were female (57.14%), had moderate ID (59.52%), and were aged 20–44 years old (71.43%). The OHK was a statistically significantly higher among participants with moderate ID than those who had severe ID at baseline in both phases (*p* = 0.035 and 0.040). There were no statistically significantly differences of PI at baseline between the control and intervention phases (all *p* < 0.05).

Table 2 shows the OHK and PI between the control and intervention phases had no difference at baseline. After board game intervention for two weeks, there was a statistically significantly improvement of OHK and decrease of PI (all *p* < 0.001), respectively. The board game intervention was effective to increase 42.29% of OHK and reduce 33.28% of PI of ID people (all *p* < 0.001).

Stepwise multiple regression analysis results in Table 3 reveal that board game intervention was the major factor responsible for the significant improvement in the OHK and reduction in PI among the participants with ID (all *p* < 0.001)

## 4. Discussion

Game-based education has been applied to children in oral hygiene courses, and it has been proven more effective than the traditional way of lecture teaching and enhances the learning motivation and effectiveness [19,20,37]. Due to the higher risk of oral health problems for people with ID, there is a gap in their proper oral health care. In addition to focusing more on providing appropriate oral health education for people with ID, some scholars also suggest people with ID learn by themselves [38,39]. In this research, OH education activities are carried out in a relaxed way in daily life through a board game which integrates OHK and skills. The results of this study indicate that oral hygiene board game intervention effectiveness improved the oral hygiene and tooth-cleaning skills of adults with ID, which in turn ameliorated their oral health knowledge and conditions and lowered their PI.

The effectiveness of OH knowledge and skills for people with ID at baseline and end point during the control period and the intervention period of this board game program showed no difference by gender or age. This is consistent with the results of Maheswari et al. [20]. Children aged 5–7 years old used the same tools and alternative OH education methods for 3 months. The OHK and lower PI results showed the play method (using a game combined with flash cards) group got significantly higher scores of 32.77% and 87.67% than the flash cards group (8.9% and 30%), respectively [20]. Although the ID people in this study are all adults, their degree of impairment is moderate to severe, and their IQs and learning ability are mostly similar to 3–6 year- old children [40]. We cannot know whether any significant relationship exists between the learning effect and the age caused by interest in the game as the research results of Maheswari et al. demonstrated [20]. However, we found the cognitive ability of people with ID is significantly related to the degree of disability [41,42]. In the pre and post-test parts of the results, it can be observed in the intervention group and the control group that people with moderate disabilities have higher oral hygiene awareness and lower dental plaque than those with severe disabilities. This shows that the learning effectiveness of knowledge and skills is still affected by the degree of obstacles among ID adults, especially in the acquisition of knowledge.

The oral hygiene score continued to improve from the first to third post-test, but the range of improvement gradually decreased. It reveals that people with ID have difficulty retaining the knowledge they obtain, i.e., they have limited short-term memory [42,43]. Especially in complex cleaning skills, people with ID lack integration ability to organize and preserve their acquired skills [44]. Furthermore, participants of this study were adults with a mean age of 37.29 years. The literature has shown that learning becomes more difficult and more time-consuming as people age because of the longer process of cognitive development to gain knowledge [45,46]. They can still learn to take care of themselves and engage in learning activities through repetitive systematic training [33,34]. In the future, related oral education activities among ID adults should be arranged twice a week in their daily schedule at the facilities to overcome the disadvantage of short-term memory and create lasting good oral hygiene practice.

Plaque removal ameliorated with time. The PI of the experimental group gradually decreased from 75.97% (at the baseline) to 62.41% (at the first post-test), 32.94% (at the second post-test), and 44.63% (at the third post-test) after the oral hygiene board game intervention. Compared with that at the pre-test, the greatest PI reduction was found at the second post-test. PI reduction at the third post-test was inferior to that at the second post-test, which may be attributed to the following two reasons: First, learning complex skills is difficult for people with ID, and they need to spend a lot of time on continuous learning and require repeated practice to acquire new skills [44,47]. Second, oral hygiene is related to habit formation, i.e., improving oral hygiene requires continuous learning, repeated practice, and long-term habit formation [48]. Overall, confirming the effectiveness of the board game intervention in reducing the PI of adults with ID should be the objective of the process. After the intervention, participants applied the brushing techniques they learnt and exhibited accurate brushing motions. Consequently, the PI significantly decreased in the post-tests.

This study has several limitations that must be taken into consideration. First, the present study was a pre-experimental study using one-group pre-test and post-test design conducted in a disabled institution, which limits the external validity of the study. However, such a research design can eliminate the heterogeneity and reduce incoordination between the two different institutions. Future studies will conduct with a rigorous study design on the period or frequency in the two groups to better assess intervention outcomes among ID adults. Second, the outcomes of this study, which were based on only the baseline results and the third time post-test results obtained after three weeks of intervention which belong to the short-term effect did not predict the follow-up duration of the effect of the oral health board game activities. It will be worthwhile to consider designing a study series to accurately verify the relationship between the intervention frequency and duration of the oral health board game program in the future. Third, we did not record in detail the OHK score improvement and the surfaces of PI reduction. Realizing the changes of OHK score and PI will help caregivers to assist the participants increase the cleaning efficiency to prevent oral diseases [9,48]. However, according to the research results, board games are effective in OH education, and there are still statistically significant positive changes in the OH status and oral cleaning skills for people with ID. There will be continuous application value in promoting the oral health of ID adults.

## 5. Conclusions

In this study, the oral hygiene changes of the participants improved after a board game-based oral hygiene education program intervention. The results indicate that the board game-based oral hygiene education program had significant short-term retention effects, proving its effectiveness for enhancing the OHK of adults with ID and reducing their PI. An oral hygiene board game was effectively used as an intervention strategy to help adults with moderate or severe ID learn about oral health related knowledge and skills.

## Figures and Tables

**Figure 1 ijerph-18-00946-f001:**
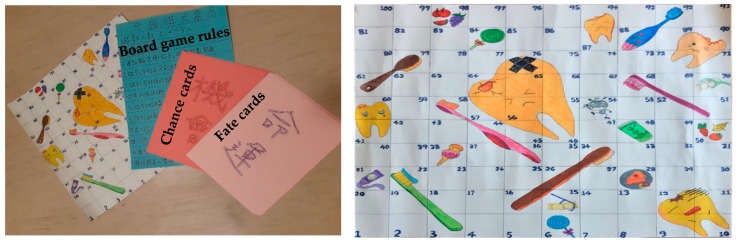
Oral hygiene board game oral hygiene board game named “Dental Monopoly” of people with intellectual disability.

**Table 1 ijerph-18-00946-t001:** Mean oral health related knowledge scores and plaque index between control and intervention phases at baseline by characteristics of participants.

Variable	N	Oral Health Related Knowledge Score						Plaque Index					
		Control Phase		*p*-Value	Intervention Phase		*p*-Value	Control Phase		*p*-Value	Intervention Phase		*p*-Value
		Mean	(SD)		Mean	(SD)		Mean	(SD)		Mean	(SD)	
Gender													
Male	18	14.44	(4.53)	0.800	13.75	(4.37)	0.848	86.94	(19.52)	0.340	80.25	(17.07)	0.323
Female	24	14.13	(3.59)		14.00	(3.49)		80.29	(23.81)		73.00	(27.98)	
Severity of disability													
Moderate	25	15.32	(3.20)	0.035	14.95	(3.06)	0.040	81.92	(22.35)	0.669	73.50	(26.45)	0.425
Severe	17	12.71	(4.55)		12.33	(4.42)		84.94	(22.21)		79.93	(20.03)	
Age group													
20–44 years old	30	14.67	(3.96)	0.302	14.32	(3.78)	0.236	82.70	(22.96)	0.840	76.04	(23.26)	0.980
≥45 years old	12	13.25	(3.98)		12.56	(3.94)		84.25	(20.61)		75.82	(27.35)	

**Table 2 ijerph-18-00946-t002:** Mean oral health related knowledge scores and plaque index between control and intervention phases at baseline and post-tests.

Variable	Baseline		*p*-Value	1st Post-Test		*p*-Value	2nd Post-Test		*p*-Value	3rd Post-Test		*p*-Value	Difference Between Pre-Test and 3rd Post-Tests		*p*-Value	% of Improvement
	Mean	(SD)		Mean	(SD)		Mean	(SD)		Mean	(SD)		Mean	(SD)		
Oral health related knowledge score																
Intervention phase	14.26	(4.08)	0.995	15.87	(3.89)	0.019	17.08	(4.15)	0.003	17.97	(4.34)	<0.001	4.31	(3.42)	<0.001	42.29
Control phase	14.26	(3.97)		13.61	(4.05)		14.23	(4.06)		13.89	(3.84)		−0.24	(0.98)		−1.34
Plaque index																
Intervention phase	75.97	(24.11)	0.166	63.03	(30.51)	0.050	34.65	(28.91)	<0.001	48.00	(30.13)	<0.001	−30.06	(30.03)	<0.001	−33.28
Control phase	83.14	(22.07)		75.94	(22.70)		77.64	(23.14)		78.78	(21.08)		−4.57	(18.58)		−1.40

**Table 3 ijerph-18-00946-t003:** Stepwise multiple regression analysis on factors associated with oral health related knowledge score and plaque index.

Variable	Term	Estimate	SE	*t* Ratio	95%CI		*p*-Value	R^2^
					Lower	Upper		
Oral health related knowledge score								
Group	Intervention group vs. Control group	4.55	0.59	7.77	3.40	5.70	<0.001	0.459
Plaque index								
Group	Intervention group vs. Control group	−25.49	5.83	−4.37	−36.91	−14.07	<0.001	0.212

Adjusted gender, age and severity of disability.

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
