# Peer review of "The Effectiveness of a Board Game-Based Oral Hygiene Education Program on Oral Hygiene Knowledge and Plaque Index of Adults with Intellectual Disability: A Pilot Study"

_ijerph, 2021, doi:10.3390/ijerph18030946_

Round 1
Reviewer 1 Report
This is a very interesting study.
I felt that it was an excellent. I felt that it was an excellent way to improve the knowledge and conditions of oral hygiene for the patients with disabilities.
If you have the opportunity for the next research, I felt that it would be good to do a comparative study on the period or frequency in the two groups.
Author Response
Response to Reviewer 1 Comments (Manuscript ID: ijerph-1070167)
Title: The effectiveness of a board game-based oral hygiene education program on oral hygiene knowledge and plaque index of adults with intellectual disability: A pilot study
Dear Reviewers,
The authors sincerely appreciate for the opportunity to consider our manuscript for publication, subject to major amendments. We have revised this paper based on the reviewer’s suggestions.
This paper has been edited by Mark Roche on January 12, 2021 and is considered to be improved in grammar, punctuation, spelling, verb usage, sentence structure, conciseness, general readability, writing style, and native English usage to the best of the editor's ability.
Your Sincerely,
The Authors
Reviewer #1:
This is a very interesting study.
I felt that it was an excellent. I felt that it was an excellent way to improve the knowledge and conditions of oral hygiene for the patients with disabilities.
If you have the opportunity for the next research, I felt that it would be good to do a comparative study on the period or frequency in the two groups.
We appreciate the suggestions. It is important to conduct with a rigorous study design on the period or frequency in the two groups to better assess intervention outcomes among ID adults. After discussed, we added the of “Future studies will conduct with a rigorous study design on the period or frequency in the two groups to better assess intervention outcomes among ID adults. “ to explain that we adopt the kind suggestions of this pilot study. (Please see Discussion in P.10 L.288-289)
Reviewer 2 Report
This work is an interesting study of practical significance. In our opinion, the introduction does not sufficiently present the idea of the study itself, so there is no general idea of the research goals and results. The experiment itself is described quite clearly, but the theoretical part of the work can be strengthened. Also, in our opinion, there is a lack of a general structure of experimental work that would allow us to trace the logic. The explanations given by the authors of the experimental work to the improvements they record are questionable. It seems to us that such changes for the better can be caused not only by experimental work, but also by other factors.
In my opinion, the work is of a practical nature and has practical significance, first of all. The fact that the study is aimed at improving people's health makes this experience very valuable. But I certainly have doubts about the scientific nature of this work. Being scientific requires a more detailed description of the approaches that researchers used, the scientific positions from which they came. There is a description of the game as a method, but there is little science in it. For me, a detailed description of all the stages of experimental work, which is thought out in advance, is very important.
1) it is necessary to introduce more scientific terminology, but its use should be justified;
2) it is necessary to show what features the game has for this category of people under study; how it differs from the use for ordinary people;
3) it is necessary to describe the sequence of steps in the implementation of the game, this will allow you to trace the logic of scientific research of the authors;
4) it is necessary to make conclusions more detailed. The idea of this research is practical, not theoretical, but when we are writing a scientific article we have to think about theoretical things first of all.
Author Response
esponse to Reviewer 2 Comments (Manuscript ID: ijerph-1070167)
Title: The effectiveness of a board game-based oral hygiene education program on oral hygiene knowledge and plaque index of adults with intellectual disability: A pilot study
Dear Reviewers,
The authors sincerely appreciate for the opportunity to consider our manuscript for publication, subject to major amendments. We have revised this paper based on the reviewer’s suggestions.
This paper has been edited by Mark Roche on January 12, 2021 and is considered to be improved in grammar, punctuation, spelling, verb usage, sentence structure, conciseness, general readability, writing style, and native English usage to the best of the editor's ability.
Your Sincerely,
The Authors
Reviewer #2:
This work is an interesting study of practical significance. In our opinion, the introduction does not sufficiently present the idea of the study itself, so there is no general idea of the research goals and results. The experiment itself is described quite clearly, but the theoretical part of the work can be strengthened. Also, in our opinion, there is a lack of a general structure of experimental work that would allow us to trace the logic. The explanations given by the authors of the experimental work to the improvements they record are questionable. It seems to us that such changes for the better can be caused not only by experimental work, but also by other factors.
In my opinion, the work is of a practical nature and has practical significance, first of all. The fact that the study is aimed at improving people's health makes this experience very valuable. But I certainly have doubts about the scientific nature of this work. Being scientific requires a more detailed description of the approaches that researchers used, the scientific positions from which they came. There is a description of the game as a method, but there is little science in it. For me, a detailed description of all the stages of experimental work, which is thought out in advance, is very important.
1) it is necessary to introduce more scientific terminology, but its use should be justified;
2) it is necessary to show what features the game has for this category of people under study; how it differs from the use for ordinary people;
3) it is necessary to describe the sequence of steps in the implementation of the game, this will allow you to trace the logic of scientific research of the authors;
4) it is necessary to make conclusions more detailed. The idea of this research is practical, not theoretical, but when we are writing a scientific article we have to think about theoretical things first of all.
The reviewer’s critiques were well taken.
- We agree that it is important to introduce more scientific terms. We used the related and specific terminology of in the dentistry education using board games. However, this is a practical research, not a pure theoretical research. We thought using popular terminology would be better than scientific terminology.
- This board game used in this study was designed according to the characteristics of cognitive ability and verbal expression among ID people. This board game differs from the ordinary board game for ordinary people. The ordinary board game usually used for fun. The board game that we developed in this study is for improving oral health knowledge and skills. The function and means of the board game for the education program is explained in the Methods
- The detailed and completed sequence of steps in the implementation of the game is described in 5.2 Implementation of the board game and 2.5.3 Board game rules.
- The authors also have revised the conclusions in the abstract and conclusion paragraphs. (Please see Discussion in 1 L.24-25 and P.10 L.307-308)
Reviewer 3 Report
Overall the article is interesting and focuses on a different approach to a problem that affects the population with it. There are some limitations inherent to this study but the authors refer to and justify them. Since this is a pilot study, it is a good start for the development of new studies and new tools to be applied to improve the oral hygiene habits of this population. Moreover, it is a reminder of such an important problem that dentists often don't care about.
I believe that the topic is interesting but it is a pilot study and, as such, there are always many things that can be improved. The authors refer to these limitations. The game is handmade - I think it would be interesting to present something that graphically is more appealing (a digital version of the board). Despite being an interesting topic, it will hardly be an article with many citations because it is not easy to reproduce the methodology and compare it with other situations. The gameplay is well explained but it lacks validity.
Specific comments:
Lines 33-37 / 38-41 / 48: The reported results relate to 2 specific studies carried out in Taiwan. That needs to be clarified.
Statistical analysis: please state the established level of significance.
Author Response
Response to Reviewer 3 Comments (Manuscript ID: ijerph-1070167)
Title: The effectiveness of a board game-based oral hygiene education program on oral hygiene knowledge and plaque index of adults with intellectual disability: A pilot study
Dear Reviewers,
The authors sincerely appreciate for the opportunity to consider our manuscript for publication, subject to major amendments. We have revised this paper based on the reviewer’s suggestions.
This paper has been edited by Mark Roche on January 12, 2021 and is considered to be improved in grammar, punctuation, spelling, verb usage, sentence structure, conciseness, general readability, writing style, and native English usage to the best of the editor's ability.
Your Sincerely,
The Authors
Reviewer #3:
Overall the article is interesting and focuses on a different approach to a problem that affects the population with it. There are some limitations inherent to this study but the authors refer to and justify them. Since this is a pilot study, it is a good start for the development of new studies and new tools to be applied to improve the oral hygiene habits of this population. Moreover, it is a reminder of such an important problem that dentists often don't care about.
I believe that the topic is interesting but it is a pilot study and, as such, there are always many things that can be improved. The authors refer to these limitations. The game is handmade - I think it would be interesting to present something that graphically is more appealing (a digital version of the board). Despite being an interesting topic, it will hardly be an article with many citations because it is not easy to reproduce the methodology and compare it with other situations. The gameplay is well explained but it lacks validity.
Specific comments:
Lines 33-37 / 38-41 / 48: The reported results relate to 2 specific studies carried out in Taiwan. That needs to be clarified.
Statistical analysis: please state the established level of significance.
- We are thankful for the suggestions. We have clarified the content of the two studies, and cited them in different parts. (Please see 1. Introduction in P.1 Line 35)
2.Accordingly, we have added a sentence for stating the established level of significance. (Please see 2.6 Statistical analysis in P.5 Line 203-204)